# O_2_ Saturation Predicted the ICU Stay of COVID-19 Patients in a Hospital at Altitude: A Low-Cost Tool for Post-Pandemic

**DOI:** 10.3390/medicina60040641

**Published:** 2024-04-17

**Authors:** Jaime Vásquez-Gómez, Lucero Gutierrez-Gutierrez, Pablo Miranda-Cuevas, Luis Ríos-Florez, Luz Casas-Condori, Marcia Gumiel, Marcelo Castillo-Retamal

**Affiliations:** 1Centro de Investigación de Estudios Avanzados del Maule (CIEAM), Universidad Católica del Maule, Talca 3460000, Chile; jvasquez@ucm.cl; 2Laboratorio de Rendimiento Humano, Universidad Católica del Maule, Talca 3460000, Chile; 3Facultad de Ciencias de la Salud, Carrera de Medicina, Universidad Privada Franz Tamayo, La Paz 4780, Bolivia; luceroguthy@gmail.com (L.G.-G.); mirandacuevasp@gmail.com (P.M.-C.); lpze.luisalberto.rios.fl@unifranz.edu.bo (L.R.-F.); lpze.luzaimara.casas.co@unifranz.edu.bo (L.C.-C.); 4Coordinación de Investigación, Universidad Privada Franz Tamayo, La Paz 4780, Bolivia; marcia.gumiel@unifranz.edu.bo; 5Departamento de Ciencias de la Actividad Física, Universidad Católica del Maule, Talca 3460000, Chile

**Keywords:** altitude, oxygen saturation, hospitals, critical care, pandemic COVID-19, adult, post-pandemic

## Abstract

*Background and Objectives*: Patients at high altitudes with COVID-19 may experience a decrease in their partial oxygen saturation (PO_2_S) levels. The objective was to assess the association between PO_2_S and intensive care unit (ICU) stay in patients at high altitudes with COVID-19. *Materials and Methods*: Clinical records of 69 COVID-19 patients (36% women) admitted to the ICU were analyzed. Median values were considered for intra-group categories (“≤11 days” and “>11 days” in the ICU) and for PO_2_S height categories (“<90%” and “≥90%”). Logistic regression and linear regression models adjusted for confounding variables were used. *Results*: Patients with >11 days in the ICU had 84% lower odds of having a PO_2_S ≥ 90% (OR: 0.16 [CI: 0.02, 0.69], *p* = 0.005) compared to those with ≤11 days in the ICU. An increase in PO_2_S by 1% reduced ICU stay by 0.22 days (β: −0.22 [CI: −0.33, −0.11], *p* < 0.001), potentially leading to a reduction of up to 1.44 days. *Conclusions*: PO_2_S is a crucial factor in estimating ICU stays for COVID-19 patients at high altitudes and serves as an accessible and cost-effective measure. It should be used in infected patients to complement the prognosis of post-pandemic ICU stay.

## 1. Introduction

One of the main challenges presented by COVID-19 was its effect on patients’ respiratory systems, potentially leading to severe respiratory distress and even death. Research indicates that a mere 1% decrease in oxygen saturation levels before hospitalization escalates the risk of death from COVID-19 by 13% [1], contributing to over 2 million deaths in Latin America [2]. Notably, mortality rates from the virus were lower in individuals residing at higher altitudes [2], likely due to the protective physiological barrier generated by high altitude conditions [3]. High altitude is defined as between 2500 and 3600 m above sea level (MASL), while extreme altitude ranges from 3600 to 5500 MASL. In these conditions, the organism produced physiological adaptations to prevent acute and chronic oxygen imbalances [4].

One such adaptation is observed in partial oxygen saturation (PO_2_S), which at high altitudes was lower in apparently healthy individuals and in patients with respiratory diseases [4]. Additionally, patients with COVID-19 may experience a decline in their PO_2_S levels due to respiratory system damage, but oxygen supplementation may be insufficient in patients infected with the virus and especially those experiencing respiratory failure [5]. The association of COVID-19 with acute respiratory disease resulted in unusually lower oxygen desaturation levels in high-altitude patients admitted to the ICU [4,6]. Furthermore, patients admitted to the ICU were 2.5 times more likely to exhibit abnormal findings on lung computed tomography scans at the 6-month follow-up [6]. 

PO_2_S exhibited more regular values in high-altitude COVID-19 survivors with ICU stay, whereas lower values were frequent in patients who subsequently died [4]. Low PO_2_S was associated with higher mortality, even at low altitudes [3]. PO_2_S was also lower in COVID-19 survivors living at high altitudes compared to those living at lower altitudes (2 to 1566 MASL) [2,3] and lower in those diagnosed with severe COVID-19 compared to those with moderate and mild COVID-19 [2].

Therefore, PO_2_S measurement became a fundamental tool for healthcare professionals in assessing the health status of patients with COVID-19 both in the ICU [1,7] and during their rehabilitation [7]. The World Health Organization (WHO) recommended PO_2_S assessment as a self-monitoring measure during the COVID-19 pandemic, considering the different skin pigmentations [5,7] for better precision, especially in conditions of low oxygen saturation [7]. Measuring PO_2_S proved to be a simple, non-invasive, and highly useful process in the public health setting, where health professionals utilized pulse oximeters to measure blood concentrations in patients. This tool could be useful in projecting the potential post-pandemic ICU stay for COVID-19-infected patients. Hence, this study aimed to evaluate the association between PO_2_S and days of ICU stay in adult patients with COVID-19 at a hospital in La Paz, Bolivia. Secondarily, the study aimed to assess its implications for the post-pandemic stage.

## 2. Materials and Methods

### 2.1. Study Design

A non-experimental, quantitative, retrospective, observational, cross-sectional, non-experimental design study. The research adhered to the recommendations of STROBE (Strengthening the Reporting of Observational Studies in Epidemiology) [8]. 

#### Phases 

The study proceeded in two distinct phases. In the first phase, we accessed and meticulously recorded patient records, ensuring comprehensive data collection. This phase was crucial for establishing the foundational dataset necessary for our analysis. Subsequently, in the second phase, we conducted rigorous statistical analyses aligned with the predefined objectives of the study. This analytical phase was instrumental in translating our research objectives into tangible results, enabling us to derive meaningful conclusions from the gathered data.

### 2.2. Participants

The sample was a purposive, or non-probability, sample. Clinical records of 69 COVID-19 patients (36% women) admitted to the ICU at a high-altitude hospital in La Paz, Bolivia, between January and September 2021 were analyzed. 

#### 2.2.1. Inclusion Criteria

Inclusion criteria were patients who had been admitted to and discharged from ICU and who had records of sociodemographic data and baseline clinical measurements. Patients lacking such information were excluded (Table 1).

#### 2.2.2. Ethical Safeguards

The study followed the guidelines of the Declaration of Helsinki, and all data were collected anonymously as part of routine hospital assessments. No informed consent signature was required, given the retrospective nature of the study [9]. Additionally, approval from the competent ethics committee, medical board, or similar authorities was not sought, consistent with another retrospective study on COVID-19 [10].

### 2.3. Clinical Investigation

The clinical records contained information on demographic variables, ICU stay, resting vital signs measured directly and indirectly, and hematological sample data. Recording of variables was done only in one measurement on admission to ICU, at a single data collection time, in order to characterize the sample at baseline. Clinical measurements were mainly systolic blood pressure, diastolic blood pressure, PO_2_S, glycaemia, partial pressure of oxygen, partial pressure of carbon dioxide, bicarbonate, hemoglobin, hematocrit, hydrogen ion concentration, red blood cells, white blood cell, protombina, sangria, capillary filling, and coagulation.

### 2.4. Statistical Analysis

Statistical analysis was conducted using STATA version 14 (StataCorp. 2015. College Station, TX, USA: StataCorp LP). Continuous variables were presented as mean values and 95% confidence intervals (CI), while categorical variables were expressed as absolute values and percentages. The Shapiro–Wilk test assessed the normal distribution of ratio variables. The Student’s t-test for independent samples or the Kruskal–Wallis test, as appropriate, examined potential differences between the sexes. The Chi–Square test (*x^2^*) or Fischer’s exact test was used to evaluate prevalence according to sex in categorical variables. To categorize ICU stay durations into two intra-group categories, namely “≤11 days” and “>11 days”, we utilized the median value (50th percentile) of the ICU stay. This approach was adopted in alignment with a previously conducted epidemiological study on COVID-19 at altitude [2], which had employed a similar percentile for classification purposes. Similarly, from the vital signs, we employed arterial oxygen saturation (PO_2_S) as a non-invasive and indirect method to establish dichotomous altitude categories. These categories were defined as “<90%” and “≥90%”. This methodological decision was informed by its applicability in previous research [11]. Logistic regression with odds ratios (OR) and linear regression with beta coefficients (β) were used to assess the association between dependent and independent variables. Various models were adjusted for confounding variables demographic (gender and age); basic anthropometrics (body weight); chronic diseases (number of co-morbidities); vital signs and semi-invasive metabolic (heart and respiratory rate, systolic and diastolic blood pressure, body temperature, Glasgow scale, pH, prothrombin, sangria, and coagulation); invasive metabolic (partial pressure of O_2_ and CO_2_, bicarbonate, hematocrit, hemoglobin, and red blood cells); and life styles (physical capacity: VO_2_max), accompanied by their respective 95% CIs. These confounding variables were chosen because in epidemiological studies the association models are adjusted for sociodemographic, anthropometric and body adiposity variables, and lifestyles. A significance level of less than 5% (*p* < 0.05) was selected.

**Table 1 medicina-60-00641-t001:** Basic demographic data and clinical variables of patients with COVID-19 in ICU at high altitude.

	Total (*n* = 69)	CI (95%)	Female (*n* = 25)	CI (95%)	Male (*n* = 44)	CI (95%)	*p*
Age (years)	56.1	52.9; 59.4	59	53.5; 64.5	54.5	50.4; 58.6	0.23
Weight (kg)	77.3	74.2; 80.4	71.1	66.2; 76	80.7	76.9; 84.4	**0.007**
Altitude (MASL)	3598.8	3511.5; 3686.1	3706	3648.1; 3763.8	3537.9	3406.2; 3669.6	**0.031**
VO_2_max (L·min)	2.11	1.93; 2.29	1.45	1.32; 1.58	2.47	2.27; 2.67	**0.0001**
VO_2_max (mL·kg·min^−1^)	26.9	25.3; 28.5	20.6	19; 22.3	30.4	28.8; 31.9	**0.0001**
*VO* _2_ *max (n, %)*							**0.012**
*Low or very low*	35	50.7	18	72	17	38.6	
*N, G, E, S*	34	49.3	7	28	27	61.4	
MET	7.5	7.1; 28.1	5.6	5; 6.3	8.6	8.2; 9.1	**0.0001**
ICU stay (days)	12.4	10.1; 14.6	12.6	9.4; 15.8	12.3	9.1; 15.4	0.611
*ICU stay (n, %)*							0.26
≤11 *days*	36	52.9	11	44	25	58.1	
>11 *days*	32	47.1	14	56	18	41.9	
Comorbidities	1.75	1.4; 2.0	1.77	1.3; 2.2	1.73	1.3; 2.1	0.861
HR (b/min)	93.2	87.6; 98.8	92.7	84.1; 101.4	93	86.1; 101.1	0.935
RR (p/min)	30.4	28.1; 32.7	28	24.4; 31.5	31.7	28.6; 34.9	0.151
SBP (mmHg)	122.4	116.1; 128.8	122.1	111.6; 132.4	122.6	114.2; 131.0	0.81
DBP (mmHg)	73.3	69.3; 77.3	74.5	68.1; 80.9	72.6	67.4; 77.9	0.724
T° (g° Celcius)	36.3	36.2; 36.5	36.4	36.1; 36.7	36.3	36.1; 36.5	0.885
PO_2_S (%)	74.9	70.4; 79.5	70	61.4; 78.6	77.7	72.3; 83.1	0.132
*O_2_S (n, %)*							0.37
<90%	50	74.6	19	79.2	31	72.1	
≥90%	17	25.4	5	20.8	12	27.9	
Glycaemia (mg/dL)	137.6	121.4; 153.7	129.8	107.6; 152	142	119.5; 164.5	0.626
PCO_2_ (mmHg)	32.1	29.4; 34.8	30.7	27; 34.4	32.8	29.1; 36.5	0.611
PO_2_ (mmHg)	66.8	60.5; 73	60.1	48.8; 71.2	70.3	62.7; 77.9	0.062
HCO_3_ (mmol/L)	21.5	20.1; 23	20.9	19; 22.9	21.9	19.9; 23.9	0.54
pH	7.4	7.3; 7.4	7.4	7.37; 7.4	7.4	7.3; 7.4	0.331
Hto (%)	47.4	46; 48.9	45	42.3; 47.6	48.9	47.2; 50.5	**0.003**
Hgb (g/dL)	15.1	14.5; 15.5	14.3	13.5; 15.2	15.4	14.8; 16.1	**0.006**
Hb glycosylated	10.3	9.4; 11.1	9.9	8.7; 11.2	10.5	9.3; 11.6	0.623
White BC	13,253.5	11,765.5; 14,741.4	12,036.5	9659.3; 14,413.7	14,033.5	12,081; 15,986.1	0.179
Red BC (mill/mcL)	5,707,395	4,129,260; 7,285,531	4,836,850	4,487,847; 5,185,853	7,354,302	3,725,914; 8,630,007	0.096
Protombina (seg)	12.4	12.2; 12.6	12.7	12.1; 13.3	12.2	12.1; 12.4	0.225
Sangria (min)	2.1	1.9; 2.1	2.1	1.9; 2.3	1.9	1.8; 2.1	**0.032**
Capillary filling (seg)	3	2.9; 3.1	3	2.9; 3.1	3.1	2.9; 3.2	0.728
Coagulation (min)	8.1	8; 8.2	8.3	8.1; 8.6	8	7.8; 8.1	**0.028**

CI: confidence interval; DBP: diastolic blood pressure; HCO_3_: bicarbonate; Hgb: hemoglobin; HR: heart rate (rest); Hto: hematocrit; ICU: intensive care unit; MET: metabolic equivalent of task; N, G, E, S: normal, good, excellent, superior; PCO_2_: partial pressure of carbon dioxide; pH: hydrogen ion concentration; PO_2_: partial pressure of oxygen; PO_2_S: partial oxygen saturation; Red BC: red blood cells; RR: respiratory rate (rest); SBP: systolic blood pressure; T°: body temperature; VO_2_max: maximum oxygen uptake (was estimated with the abbreviated method of Wasserman et al. [12] and categorized according to López and Fernández [13]); White BC: White blood cell (count). Source: own elaboration.

## 3. Results

The sample of COVID-19 patients at high-altitude conditions hospitalized in the ICU showed differences by sex for body weight and altitude of residence. Specifically, body weight was higher in men, while the altitude of residence was higher in women. Additionally, distinctions were observed in certain clinical variables, including hematological samples, such as hematocrit and hemoglobin. In this regard, levels were higher in men. Moreover, non-invasive measurements, such as bleeding and coagulation, indicated variations, with higher values recorded in women (Table 1).

The median length of stay in the ICU at high altitude was 11 days, corresponding to the 50th percentile. Consequently, patients with a COVID-19 ICU stay exceeding 11 days exhibited an 84% lower likelihood of having a PO_2_S equal to or greater than 90% compared to those with stays of 11 days or fewer in the ICU (OR: 0.16 [CI: 0.02, 0.69], *p* = 0.005). When adjusting for confounding variables, all models indicated reduced odds of increased PO_2_S, except model 4. Of these, model 3 had the lowest odds (95%) of elevated PO_2_S at altitude (Table 2).

When associating the variables of interest, we found a significant association wherein a 1% increase in partial oxygen saturation (PO_2_S) resulted in a noteworthy reduction of 0.22 days in intensive care unit ICU stay for high-altitude COVID-19 cases (Table 3, model 1). Consequently, if PO_2_S were to increase from the lowest recorded value of 26% to the highest observed in the sample at 98%, the ICU stay at altitude would decrease by 1.66 days. This trend was consistent across adjusted models, with models 2, 3, and 5 also indicating reductions in ICU stay associated with a PO_2_S increase from 26% to 98% (Table 3).

## 4. Discussion

The study aimed to assess the association between PO_2_S and days of ICU stay in patients with COVID-19 at high altitudes and its projection in the post-pandemic stage. The key finding was a significant correlation between higher PO_2_S and a reduction in ICU stay duration for COVID-19 patients at high altitudes.

Comparative evidence indicated that COVID-19 survivors residing at higher altitudes had lower resting PO_2_S compared to those at lower altitudes [2]. PO_2_S values of 92–93% were found in high-altitude COVID-19 survivors [2], equal to 94.2% (±7.9) in COVID-19 patients [10]. PO_2_S values ≥ 85% were also reported in a large number of high-altitude COVID-19 hospitalisations [3], a median of 90% (86–93) [14], an average of 88.2% (±3.3) [15], and a median of 90.5% (86.9–93.6) in patients in ICU for COVID-19 at high altitude, the latter being lower in those who died than in survivors [16]. With all this evidence, PO_2_S of between 89% and 93% was established as normative values at altitude, increasing patients’ chances of COVID-19 survival [4].

These recently reported findings indicated to us that PO_2_S values in high-altitude regions are relatively similar or stable. Our results were somewhat mixed, showing an average PO_2_S equal to 74.9%, compared to approximately 90% in the evidence presented in the previous paragraph. For their part, the PO_2_S values we reported in our study (74.9%) were closer to an investigation where patients hospitalized for COVID-19 at altitude with a PO_2_S of 75–79% and ≤75% had 2.92 (1.99, 4.3) and 2.79 (1.89, 4.11) (both *p* < 0.001) higher risk of ICU admission or death, respectively, compared to patients with PO_2_S ≥ 85% [3]. Somehow, the literature confirmed these postulates by stating that patients with COVID-19 hospitalized in ICU at high altitudes (3339 MASL) with higher oxygen respiratory rates had lower odds of mortality [17]. Patients hospitalized at low altitudes with a PO_2_S ≤ 90% were 47 times more likely to die due to COVID-19 than those with a PO_2_S > 90% (*p* < 0.001) [18].

Women survivors of COVID-19 in high-altitude ICUs (≈3500 to 4150 MASL) had a length of stay of 20.1 days, while in men it was 15.9 days [4]. In adults with COVID-19 admitted to a high-altitude ICUs (≈2640 MASL), an average length of stay of 7 days (range 4–13) was reported [6]. Other studies found a median ICU stay of 9 days (inter-quartile range 4–15) at high altitudes (≈2640 MASL) being longer in patients with mechanical ventilation than without ventilation and longer in those who died compared to survivors [19]. A median ICU stay of 8 days (inter-quartile range 3–15) at altitude for COVID-19 was reported, being longer in those who died compared to survivors [16], and a median hospital stay for COVID-19 at altitude (≈3800 MASL) of only 3.8 days [3].

The literature pointed out that the length of stay in health centers varied and was very different in high altitude conditions, probably explained by the adverse events that patients may have suffered, such as mortality, which occurred unexpectedly due to the pandemic characteristics of COVID-19 (availability of beds, mechanical ventilators, unfamiliar and unexpected situations, etc.). Our results indicated an ICU stay of 12.4 days for the total sample, which was within the range of approximately 3 to 20 days according to the data presented in the preceding paragraph. Our records were also similar to an investigation in which a stay of 11.3 days associated with the mortality event was reported [4], although no deaths occurred in the present investigation up to the date of data collection.

Regarding this last issue of mortality, we might think that the patients whose medical records were analyzed and who were part of the present study could probably not have suffered any adverse events after the data collection period, i.e., no mortality was reported. This assumption should be considered because, on the one hand, PO_2_S monitoring has been associated with lower mortality in patients with COVID-19 compared with patients in whom no monitoring instruments were used [7], and, on the other hand, COVID-19 mortality has been lower in higher altitude regions (≈2133 MASL) compared with lower altitude regions (≈ <914 MASL) [20]. For these reasons, it is important to survey PO_2_S measurement in COVID-19-infected patients admitted to ICU in high altitude regions, as evidenced in the present study, which is certainly relevant to be transferable from its practical utility to the current post-pandemic stage, i.e., to project the valuable prognostic value of PO_2_S in COVID-19 patients at high altitude. In this way, we provide important evidence in favor of the secondary objective we have set out in this research.

One potential limitation of this research was the cross-sectional nature of the study, which did not allow us to assume a cause-and-effect phenomenon for the results found, even though statistical significance was achieved in the association models. Therefore, such results should be taken with caution. However, this research had strengths that should be highlighted, one of them being that the use of clinical records was made available during the COVID-19 pandemic period, which allowed working as objectively as possible with patient data, and also that the clinical records included objective evaluations and procedures for collecting data on the variables analyzed.

## 5. Conclusions

We conclude that PO_2_S measurement is a crucial indicator for estimating the ICU stay in COVID-19 patients in high-altitude conditions. We have demonstrated that PO_2_S is an accessible and cost-effective measure, carrying significant implications for public health during the COVID-19 pandemic. These medical and practical implications are relevant in the occupational field; therefore, it is recommended that, post-pandemic, PO_2_S measurement continues to be employed to complement the prognosis of ICU length of stay, emphasizing its enduring relevance as a valuable tool in the healthcare management of individuals affected by COVID-19. 

The implications of this study extend beyond its immediate findings and could be projected into various areas of research. One potential avenue is to investigate the association between PO_2_S and lung capacities and volumes in patients infected with COVID-19. Given that lung function parameters are crucial indicators of respiratory health, understanding their relationship with PO_2_S could provide valuable insights into the respiratory consequences of COVID-19.

Furthermore, exploring the association between PO_2_S and the capacity to perform aerobic physical exercise at altitude in COVID-19 patients could yield important clinical implications. As physical exercise is integral to overall health and well-being, particularly in the context of recovery from respiratory illnesses, understanding how PO_2_S influences exercise tolerance could inform rehabilitation strategies for COVID-19 survivors. 

## Figures and Tables

**Table 2 medicina-60-00641-t002:** Probability of oxygen saturation of patients with COVID-19 in ICU at high altitude.

PO_2_S ≥ 90%	OR	*p*-Value	CI (95%)
Model 1	0.16	**0.005**	0.02; 0.69
Model 2	0.15	**0.008**	0.03; 0.61
Model 3	0.05	**0.017**	0.0; 0.6
Model 4	1.08	0.95	0.09; 12.7
Model 5	0.07	**0.038**	0.01; 0.86

CI: confidence interval (lower and upper limit); OR: odds ratio. Model 1 unadjusted. Model 2 adjusted for sex and VO_2_max. Model 3 adjusted for age, weight, and number of comorbidities. Model 4 adjusted for heart and respiratory rate, systolic and diastolic blood pressure, body temperature, Glasgow scale, pH, prothrombin, sangria, and coagulation. Model 5 adjusted for partial pressure of O_2_ and CO_2_, bicarbonate, hematocrit, hemoglobin, and red blood cells. Models 3, 4, and 5 were further adjusted for sex and VO_2_max. Source: own elaboration.

**Table 3 medicina-60-00641-t003:** Association between ICU stay and PO_2_S of patients with COVID-19 in high-altitude condition.

ICU Stay (days)	Constant	β	*p*-Value	CI (95%)
Model 1	28.9	−0.22	**<0.001**	−0.33; −0.11
Model 2	26.2	−0.24	**<0.001**	−0.35; −0.12
Model 3	36.4	−0.3	**0.001**	−0.46; −0.14
Model 4	−33.3	−0.42	**<0.001**	−0.63; −0.22
Model 5	39.8	−0.27	**0.002**	−0.44; −0.11

CI: confidence interval (lower and upper limit). Model 1 unadjusted. Model 2 adjusted for sex and VO_2_max. Model 3 adjusted for age, weight, and number of comorbidities. Model 4 adjusted for heart and respiratory rate, systolic and diastolic blood pressure, body temperature, Glasgow scale, pH, prothrombin, sangria, and coagulation. Model 5 adjusted for partial pressure of O_2_ and CO_2_, bicarbonate, hematocrit, hemoglobin, and red blood cells. Models 3, 4 and 5 were further adjusted for sex and VO_2_max. Source: own elaboration.

## Data Availability

The data is unavailable due to privacy or ethical restrictions.

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
