# Peer review of "O2 Saturation Predicted the ICU Stay of COVID-19 Patients in a Hospital at Altitude: A Low-Cost Tool for Post-Pandemic"

_medicina, 2024, doi:10.3390/medicina60040641_

Round 1

Reviewer 1 Report

Comments and Suggestions for Authors

The article with the title - O2 Saturation Predicted the ICU Stay of COVID-19 Patients in a 2 Hospital at Altitude: A Low-Cost Tool for Post-Pandemic, is interesting and well structured.

To increase the relevance of the study, we make the following recommendations:

Introduction - To detail the new aspects in relation to previous studies that address the same topic.

Material and Methods - Recommend restructuring the content into subsections: 2.1. Study design with emphasis on research phasing; 2.2. Participants - with mention of the inclusion criteria in the study; 2.3. Clinical investigation / record; 2.4. Statistical analysis.

Conclusions: identification of the medical and practical implications in relation to the most relevant results of the study. Identification of future research directions related to the results of this study.

Author Response

Dear reviewer, 

Please review the attached document.

Thank you.

Regards. 

Reviewer 2 Report

Comments and Suggestions for Authors

I appreciate the authors for presenting this clinically useful research article, which emphasizes the important role of partial oxygen saturation (PO2S) in COVID-19 patients at high altitudes. This is a well-designed and organized research article. Based on the provided report, PO2S appears to be a significant prognostic factor for COVID-19 patients, particularly in the context of ICU stays at high altitudes. The study found a significant association between higher PO2S levels and a reduction in ICU stay duration for COVID-19 patients at high altitudes. Since oxygen levels may already be lower at high altitudes due to environmental factors, maintaining adequate oxygen saturation becomes particularly important. Therefore, PO2S monitoring may carry heightened significance in such settings, as hypoxemia can exacerbate respiratory distress and lead to poorer outcomes. My comments are as follows:

 1. Prognostic factors such as disease severity, inflammatory markers, and treatment interventions could also play critical roles in determining patient outcomes. Why did the authors choose to focus solely on these parameters for study?

 2. The rationale for parameter selection within each model (model 2 to 5) should be explained in the methodology section.

 3. In the case of prolonged patient stays in the ICU, how were the parameters selected and subsequently analyzed? Was the statistical analysis based on the aggregation of data from all days, or was the analysis conducted on the worst-case data selected from each day? Please clarify this in the methodology section.

Author Response

(The authors gave the same response as above.)

Round 2

Reviewer 1 Report

Comments and Suggestions for Authors

The authors improved the manuscript acoording with the recomandations.